# Architectural Competitions on Aging in Denmark Spatial Prototypes to Achieve Homelikeness 1899–2012

Jonas E. Andersson 

Faculty of Culture and Society, Department of Urban Studies, Malmö University, SE-20506 Malmö, Sweden; jonas.andersson@mau.se

**Abstract:** In Denmark, appropriate architecture for aging is an engaging topic, often explored through the use of architectural competitions. Since 2013, national guidelines for homelike architecture for eldercare have been in place, open for use in contemporaneous competitions. This study is focused on architectural competitions prior to 2013 and the development of modern architecture for aging. Based on reports on competitions in professional publications for architects, this study covers the period of 1899–2012. Inspired by the French philosopher Paul Ricoeur's view of architecture as a spatial practice that 'does not invoke what no longer is there but what has become through what is no longer present', the present study revisits competitions on architecture for aging in search of inspirational input and links to the national socio-political discussion. This study uses case study methodology with a mixed method approach. A total of 76 competitions are identified, mainly organized by Danish municipalities, and are linked to four paradigms in social legislations. It is concluded that early competitions defined spatial prototypes, both for the homelike setting and the institutional environment, which have been continuously revisited. Since 2008, homelikeness has become the main design criterion for architecture for the frail aging population with an increasing dependency on caregiving.

**Keywords:** architectural competitions; aging; socio-politics; spatial prototypes; homelike setting; institution-like environment



## 1. Introduction

Given the slower pace of change in architecture, buildings may solidify cultural and societal beliefs and reflect past or present aesthetic, ethical, or political ideas [1–3]. Cultural and social ideals nourish theories on the social aspect of the built environment and thereby influence architectural designs [4–6]. For instance, hospitals, kindergartens, schools, and other types of societal institutions are components of the emerging modern society [7–9]. In this context, architectural competitions often act as an experimental arena for aesthetic, ethical, and functional dialogue regarding welfare goals, spatial needs, and functional requirements [10]. The explicit purpose of most architectural competitions remains the same over time, i.e., to innovate and rethink existing architectural practice and spatial usages [11,12]. Among member states of the European Union, competitions define the first step of a tendering process for a new building in which the competition jury awards the subsequent commission to the perceived best proposal [13,14]. In addition, the first prize winner receives a sum of money. Other participants, such as second or third prize winners, may also receive such sums. These sums are presented in the invitation or open call to participate in the competition. The competition jury may also assign honorary mentions, at best associated with a low prize sum. Architectural competitions contain three main phases [15]:

- The organizer's formulation of the design task and the competition brief with the client's/organizer's vision for the new building, physical requirements, and other material;

- The participating architects' interpretations of the brief into architectonic visions; the proposals of the participating architects' interpretations of the competition brief with requirements in terms of architectonic visions for buildings and urban space;
- The jury's evaluation of the submitted proposals to designate a winner; the jury assessment report with evaluations of the submitted proposals and their feasibility in relation to the intentions of the competition.

In adopting a design theoretical approach for architectural competitions, semantically value-charged keywords serve as kernels for creating mental figures—generating images—which guide and structure the architects' works in a certain direction by combining fragments of other types of architecture into new spatial prototypes [16,17]. Their knowledge of human behaviors and building usages becomes an intrinsic part of their creative work in interpreting a competition brief. In an iterative procedure interrupted by formative decisions based on spatial and technical requirements, textual-bound information merges into three-dimensional manifestations and a final architectural design [18–20]. The finalized building can be described as the harmonization of the clients' intentions with the artistic ambitions of the winning architects [21]. Winning proposals constitute a bank of referential prototypes for other types of architecture, active for a long period and creating either imaginary cul-de-sacs or paths to new ideas for the corpus of architects [18,22]. In Denmark, the national professional organization for trained architects, the so-called *Akademisk Arkitektforening* (Association for Danish Architects, hereafter referred to as ADA), has published articles about architectural competitions with varying design tasks since 1898. As in other European countries, the ADA has controlled the architectural competition as a professional instrument for promoting high building quality since the mid-nineteenth century. As the ultimate evidence of being part of a specially educated corpus of professionals in architecture [19], it provides unique insight into the aesthetic, ethical, and functional reasoning among architects, even surpassing corresponding publications in Norway and Sweden. Following the European competition tradition, competing in architecture is not simply about landing a new commission; rather, it is a quest for the professional recognition of being a talented and genial architect [19]—to '*suppress the ignoramuses*' and '*find the ultimate spatial solution*' [23].

*Nordic Architectural Competitions on Buildings for Societal Use*

In European countries, competitions tend to produce inspirational models that the corpus of architects may refer to, reuse, and reinvent for the same usage or other purposes. Awarded proposals and honorary mentions form a type of architecture that generates spatial prototypes for specific functions or new purposes. Paradigms in architecture can be retraced through architectural competitions [24–27]. Norwegian and Swedish research suggests that residential architecture for the aging population can be associated with the emergence of the modern Nordic welfare state in the late 19th century and its continued development during the 20th century [26,27]. Among these countries and Denmark, a link can be found between social reforms concerning aging and retirement and architectural competitions. Social–political visions were converted into spatial requirements present in competition briefs and tested as design criteria for the participating architects' proposals [24–27]. Today, both the Norwegian and the Swedish welfare models promote the homelike setting as beneficial for the frail aging process rather than institution-like settings [26,27].

Denmark has organized architectural competitions concerning appropriate architecture for the frail aging process. Common for this type of architecture is the alignment of architectural features (i.e., the level of a perceived homelike setting from the older person's perspective) with work–environmental requirements and care hygiene regulations (i.e., the level of the presence of institution-like features). To follow the national research project on appropriate housing for older people, or the 'Well-being and housing' project 2004–2008, Danish municipalities, the Danish Energy Agency, and the private fund REALDANIA initiated a project called 'Societal institutions for the future' in 2009 [28,29]. This project

resulted in detailed architectural and functional requirements for kindergartens, schools, and care institutions that must be realized in the architectural design for such buildings, which are accessible on a special website [30]. These guidelines are mandatory since the Danish state subsidizes approximately 3–5% of the building costs for such buildings. Since 2013, these requirements have often been used as competition briefs for new architecture for the frail aging population or as an appendix to purpose-composed competition briefs.

Inspired by design theorist and French philosopher Paul Ricoeur's view of architecture as a spatial practice that 'does not invoke what no longer is there but what has become through what is no longer present' [31,32], the aim of the present study is twofold: firstly, to study the presence of underlying generator images in architecture for older people during the period of 1899–2012, and secondly, to use a transdisciplinary approach so that architectural competitions may be linked to the socio-political discussion about aging. With this double objective, the intention is to retrace the generating images that were activated during the evolution of housing for frail older people in the modern Danish welfare state.

Case studies are often essential to developing new knowledge about a complex phenomenon [33]. It is surmised that the ADA journal entitled 'The architect' in the Journal of the Association of Danish Architects (hereafter referred to as JADA) published in different formats over the years, either as a weekly, monthly, or yearly issue, may shed light on architectural competitions that focused on architecture for aging. This material is accessible at the library of the Royal Academy for Fine Arts in Copenhagen. The working hypothesis of this study is twofold. Firstly, it is hypothesized that socio-political ideals from different stages of the formation of the Danish social act are reflected in architectural competitions in a similar manner as in neighboring countries. Secondly, it is hypothesized that the submitted proposals encapsulate veins of the ongoing socio-political discussion about appropriate architecture for aging.

## 2. Material and Methods

The present study was a multiple case study that adopted a type of spatial hermeneutic approach to suit its Ricoeurian inspiration and combined research on architecture with research on eldercare and social welfare [31,32,34]. The research material was assembled via a close reading process of publications from the ADA. This reading primarily helped to identify architectural competitions from the research period; in addition, by use of concepts or keywords, it also situated the competitions in socio-political contexts that could be further explored in secondary sources or research on social welfare. This secondary material consisted of publications from research in social welfare or public white papers. The JADA was complemented by three ADA compilations on architectural competitions in the period 1907–2001 [35–37]. Competitions in the period 2002–2012 were assembled by studying the ADA website.

The study used triangulating research methods, i.e., close reading of the JADA and additional documents, archival research with keywords, analyses of architectural drawings (mainly floorplans), and study visits [34]. Architectural competitions with a specific focus on architecture for aging were found by scrutinizing the JADA, and each incident constituted an individual case. Each case was subjected to close reading to extract further information about the competition [38]. Documents, events, or legal reforms mentioned in articles in the JADA were checked for additional information that could shed light on the motives for the specific competitions. The national library database was used to check for authors, publications, and concepts. It contains legal deposits of at least one copy of every publication made in Denmark since the 17th century. However, searches using the words 'architectural competition' or 'old people's home' in Danish and English generated poor results. Information about locations and addresses was checked through public registries and digital maps mainly supplied by Google Earth. This allowed for controlling whether the competition had resulted in a new building, a new settlement, or ascertaining that the building was probably torn down. Public records and websites were consulted to deduce current activities and prepare for later study visits.

## 3. Results

The research allowed for identifying 76 architectural competitions with a specific focus on appropriate architecture for aging during the period of 1899 to 2013, as is shown in Figure 1. This period could probably have been extended further backwards for two reasons: Firstly, the JADA did not introduce a regular column on architectural competitions (forthcoming, pending, and finalized competitions) until 1907. Secondly, Swedish studies on aging mention some earlier Danish competitions concerning housing for older people [39–41]. The research material consisted mainly of winning proposals since the JADA initially focused its competition reports on the first prize winner but gradually included second and third prize winners or honorary mentions. The year 2013 represented a sharp change in the formulation of competition briefs on architecture for the frail aging. Until then, most competitions adhered to the general principles for architectural competitions, i.e., competition organizers formulate specific competition briefs for an individual competition task.

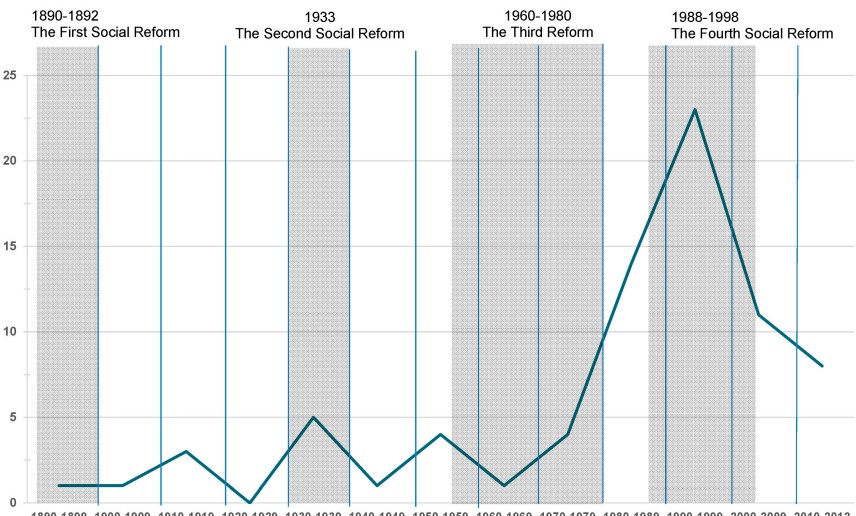

**Figure 1.** The 76 architectural competitions identified in the JADA that focused on architecture for housing for older persons during the period 1899–2012 and the four reforms of the Danish social act (light-gray fields) (figure by author).

The majority of the studied competitions (62 competitions) were invited competitions with prequalification procedures for selecting participating architects or for inviting the organizers' preselection of 3–5 architectural firms. A common trait among most of the competitions was that the winning architects received a prize sum and subsequent commission. In most cases, lower prize sums were issued for second and third prize winners. A small number of competitions, about 14 competitions, were so-called open competitions for trained architects, i.e., members of the ADA, without any prequalification or participant selection processes. A clear majority of the competitions were organized by public players. Thus, Danish municipalities organized most of the competitions, i.e., 54 competitions, and in some cases, larger municipalities organized more than one competition for the period. Five competitions were organized by a consortium of interested partners, being either a municipality, county, or ministry—all three are part of the Danish civil administration—in close collaboration with one or several housing cooperatives. Individual housing cooperatives organized 14 competitions.

### 3.1. An Architectural Typology of Space for Aging

The peaks in the number of competitions suggested a correlation with the development of the Danish welfare state often described as a four-step process with distinct paradigms in the political discussion on social welfare [42,43] (see Figure 1). The 76 architectural competitions generated spatial prototypes for architecture for aging that placed increasingly

greater attention on the older person and his/her needs (see Figure 2). During the period of 1899–2012, these prototypes were revisited, dissected, and reassembled in attempts to complement ongoing socio-political discussions about the beneficiaries and boundaries of the national social act. Together, the prototypes formed an architectural typology for aging that functioned as a capital of generating images for converting socio-political reforms into an envisioned type of architecture for aging. By the end of the 20th century, the prototypes merged into a contemporaneous prototype for residential care homes and eldercare centers (see Figure 2).

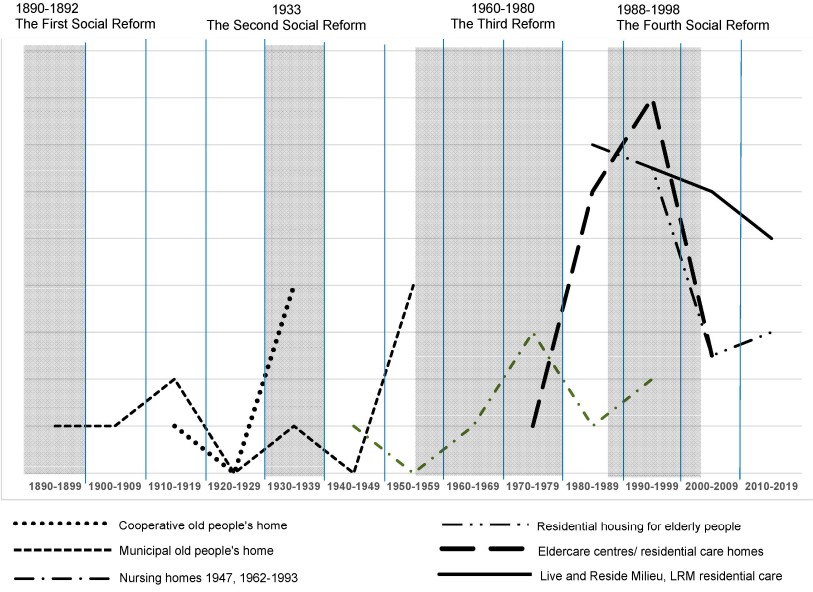

**Figure 2.** Overview of seven spatial prototypes that appeared in architectural competitions of 1899–2012 in answer to social reforms (figure by author).

The slower pace of change of architecture made these prototypes into three-dimensional concretizations of beliefs about the ideal space for aging that could be further interpreted by studying research on social welfare. Consequently, the prototypes could be closely linked to cultural beliefs about aging which permeated the textual documentation of the competitions. The underlying ethical and spatial discussion found in the competition documentation or articles in the JADA revolved around whether the aging process could be considered as an undignified type of aging that burdened and strained the surrounding society or whether it was a type of dignified aging that called for esteem and social respect in society [42,43]. A theorem for understanding how the architectural competitions from the period of 1899–2012 acted as generators of spatial prototypes that have forged contemporaneous views on the appropriate space for aging can be formulated. This theorem suggests that ethical views on aging activated two main generating images that influenced the production of spatial prototypes for housing older people: either the everyday homelike setting or the regulated institution-like environment.

### 3.2. Architectural Competitions Related to Four Paradigms in Social Legislation

The following four sections present the 76 architectural competitions in relation to paradigms in the development of social legislation. The sections argue that the architectural competitions on aging for the period 1899–2012 generated spatial prototypes that correlated with socio-political discussions about the construction of the Danish welfare state.

#### 3.2.1. Competitions 1899–1929: Prototypes for Small Pension Housing

During the 1880s, the then recently unified Germany aimed for domestic stability by introducing social reforms for the working population [44,45]. Thus, a health insurance was passed in 1883, followed by accident insurance in 1884 and old-age and disability insurance

in 1889. The German social model had a strong influence on Danish social reforms [44,45]. A decade later, Denmark started a development into becoming a modern welfare regime inspired by the fin-de-siècle conservative–cooperative movement of pre-industrial society together with liberal and social-democratic ideas to reform society [44,46]. What is called the *First Social Reform* of 1890–1891 established a poor relief system that improved the existing sickness insurance system. The reform introduced municipal help funds that targeted people with functional disabilities, long-term medical conditions, old age, or victims of other unforeseen events [45,46]. The system was implemented differently in rural and urban municipalities, mainly depending on the level of social-democratic influence on local politics [47]. In larger cities, the implementation converted the old system into an early welfare regime that was open to every citizen and with similarities to the present social system [48]. In 1891, the Danish parliament passed a new law on a regular small pension for another group of older people aged above 60 years who had not previously required municipal help but due to frailty were at risk of becoming beneficiaries of poor relief aid [45]. In total, the ratio of older people aged 60 years and older was 2.8% of a population of approximately 2 million people (Statistics Denmark).

A Municipal Prototype Focused on Accommodation

In 1899, the municipality of Aarhus initiated an open architectural competition for a new old people's home as what was called 'an asylum for people with old age pensions' [49]. The competition site was next to the city's burial ground. The competition brief stipulated that the building should have approximately 100 rooms that were to be shared by 1–2 residents. The building also had to accommodate 30 residents in a special hospital ward. In addition, the interior space should have ample daylight and appropriate ventilation. Eighteen proposals were submitted to the competition [50]. In 1900, the competition jury awarded a proposal designed by architect V. Schmidt as the one best fulfilling the list of requirements [50]. The JADA omitted the motto for the winning proposal as well as wherever other awards or honorary mentions were issued. Construction works began in 1901. At the inauguration ceremony that year, an influential social-democratic member of the city council declared [48]: 'The true essence of humanity has now also come to the city of Aarhus' [48]. See Illustration S1 (in supplementary materials), new scan of article in 300 dpi [51].

In 1904, the JADA gave the winning architect the possibility to present the new building [49]. The floorplan showed a C-shaped building with a central long corridor system around which the residents' rooms were organized. In the center of the floorplan, there was a communal space for meals, chores, and religious and social events (see Illustration S1). The drawings indicate a new central heating and ventilation system that would ensure a healthy indoor climate [49], probably a calorifier system that used cooled or heated air to regulate the indoor climate and remove stale air. In contrast to other asylums or homes for older people within the municipal poor relief aid, e.g., the newly opened municipal asylum for older people in Copenhagen designed in 1897 as a regular commission, the new housing in Aarhus only had shared rooms for a maximum of two residents. However, the configuration of the Aarhus project shared several design features discernable in the design for municipal asylums. The final building accommodated 141 older people in 70 shared rooms. In addition, 11 single rooms were available. Each floor had separate but shared space for bathrooms and water closets placed close to the central staircase. The hospital ward had approximately 40 beds [49]. Staff members were housed in the basement or in the attic. The building is still standing today, but a new wing has been added and an enclosed courtyard created.

In 1907, a call for a second architectural competition on housing for a new old people's home in the municipality of Svendborg was published in the JADA [52]. It was an invited competition for three pre-selected architects. Instead of a jury, a state representative of the Royal Building Inspection Board assessed the submitted proposals [53]. Only the winning proposal, which depicted a two-story building in red brick intended for 60 older persons,

was given an award [53]. The residents were accommodated in individual or two-person rooms [52]. The floorplan had a C-shaped configuration with a central corridor system around which the rooms were organized. The corridors led to communal spaces for meals, chores, and religious and social events, which were situated at the end of the corridor, thus making the corridors appear shorter. All the rooms had shared bathrooms and toilets. The building was equipped with technical installations such as an individual heating plant and a food elevator from the central kitchen connecting each floor [53]. The building also included a hospital ward with 20 beds [53]. The building exists today for mostly the same use, although it has been refurbished to fit modern requirements for residential care homes. See Illustration S2 (in supplementary materials), new scan of article in 300 dpi [54].

Between 1912 and 1914, a third architectural competition about an old people's home was held in the municipality of Naestved and presented by the JADA in 1917 [55]. In this case, three architects submitted proposals to the city council in response to the council's commission of new housing for a group of old persons with pensions by a fourth architect [55]. The ad hoc competition forced the council to seek outside expertise to evaluate the proposals. In the end, the council chose the contracted architect's proposal, and the other proposals were set aside. The housing was intended for 35 residents, but it did not include a hospital ward. The floor plan had mainly one-person rooms but also some two-person rooms, all with shared bathrooms and toilets. In this case, the floorplan had a central corridor that ended in a communal space for meals, chores, and religious and social events. This building is still standing and has the same use today. The center accommodates approximately the same number of residents, but extra square meters have been added to improve space for dining and social activities. The center also includes space for rehabilitation and physiotherapy.

A fourth architectural competition about a new old people's home was organized in 1919. It was an open competition in the municipality of a small town in the Copenhagen area [54]. In this competition, the jury awarded three proposals with first, second, or third prize. All three awarded proposals repeated spatial prototypes from previous competitions for this type of housing, e.g., a central corridor system connecting two-person shared rooms and one-person rooms with a central space for dining, chores, and social activities (see Illustration S2). The proposal had mainly individual rooms with shared facilities for hygiene. One floor contained work-related functions and housing for the staff. The first prize winning proposal was not realized. Instead, it was the second prize winning proposal that was built, albeit with many similarities to the first prize proposal. This building remains standing for its original use, but it has been extended and refurbished.

A Cooperative Prototype with Residential Qualities

As early as 1896, the denomination 'old people's home' was used for a housing cooperative project for retired postmen in Copenhagen. This was a regular commission. The first architectural competition organized by a housing cooperative occurred some sixteen years later. In 1912, the Association for Apprenticeship Training in Arts and Crafts called upon architects to participate in an open architectural competition for a new housing estate at the fringe of a green area in the northern parts of Copenhagen. The brief envisioned a building to be realized in defined phases over a longer period [56]. It stipulated 120–125 one-room flats for individual older persons living alone and approximately 18 one-and-a-half-room flats for married couples. The flats were to have a hallway with storage facilities, a kitchen, and living-room space of fourteen square meters. Five square meters were added for an alcove with a double bed for married couples. The building was to be equipped with electrical lighting, gas for cooking, and a central water-heating system. Space for hygiene, a laundry, and WC was to be shared by the residents and located on the basement floor. Elevators for three users were to be installed in stairwells that served flats intended for frailer older persons [56]. See Illustration S3 (in supplementary materials), new scan of article in 300 dpi [57].

The competition generated great interest among architects. By 1913, the association had received 40 proposals. An eight-member jury evaluated the architectural designs in relation to what the jury deemed as the most essential quality for architecture intended for aging well: the pleasant sensation of being at ease and at home [57]. Other assessment criteria were ample daylight penetration, healthy aeration, and low building costs. The competition was thoroughly documented in the JADA in a ten-page article [57]. The jury concluded that no individual proposal had achieved all the assessment criteria. Thus, the prize sum was split equally between four proposals [57]. All the awarded proposals contained a large-scale building with a total of 145 flats with interior courtyards for social activities facing the park. The proposals contained one-person flats of about sixteen square meters and flats for married couples of about twenty square meters.

The jury's decision caused great dismay among architects and the ADA. A heated debate followed in the JADA about national rules for architectural competitions, prizes, honorary mentions, and winning proposals. The JADA lobbied to sue the jury, but this never occurred. The decision also caused the association problems in choosing which proposal to realize, which was completed two years later (see Illustration S3). The full proposal was never entirely built but was instead only partly built alongside the street without the envisioned expansion facing the park. The housing estate was inaugurated in 1915. The building is still standing with the proud name 'Haven of peace-home for workers', but its residents belong to younger age groups than initially intended.

Conclusions Pertaining to Competitions 1899–1929

During the period of 1913–1919, the JADA published several articles about appropriate architectural designs for homes intended for different users, therein including furniture and interior decorating. The essence of the home environment for various groups was also explored in articles. The articles reflected ongoing events, such as the Scandinavian conference on improved quality in housing for people with smaller incomes, which opened in 1917. A year later, a special exhibition on social issues opened, in which architecture, furniture, and technical installations were displayed as instruments for solving the social imbalance in society.

WWI put an end to further architectural competitions on appropriate architecture for aging. Instead, the JADA presented other architectural competitions with public functions such as churches, town halls, schools, and public places [35]. Wartime conditions caused severe shortages of building materials and the construction of new housing declined to a minimum. To boost the slumbering building market, the government introduced subsidies for the construction of new housing [58]. Over the period of 1917–1921, housing cooperatives participated in increased building activity that focused on solving the housing shortage and supplying new solutions for cooperative dwellings [58].

3.2.2. Competitions 1930–1959: A New Take on Architecture for Aging

The general election in 1929 led to the social democratic party forming government in collaboration with the social-liberal party. The *Second Social Reform* occurred in 1933. In 1930, the ratio of people aged 65 years and older was about 7.3% of a population of 3.5 million people (Statistics Denmark). The reform introduced four new laws that expanded the fin-de-siècle social system [46]. A new law on public assistance replaced the old system of public funds of 1891–1892. A new public insurance law integrated the existing sickness insurance law and the old pension law with a new extension that targeted invalidity. In addition, public support was extended to include insurance in case of work-related accidents, unemployment, or being assigned to take part in public works [46]. The reform had a significant impact on the working population, and as such, it is often referred to as the great social reform. After WWII, the social-democratic take on defining the modern welfare society continued along with the realization of the reform of 1933 that had been postponed by the war. A smaller social reform was realized in 1956.

Municipal Prototypes Targeting Accommodation

The reform of 1933 enforced the provision of old people's homes by municipalities [46,59]. The reform also prepared for developing hospital wards into a new building type that was intended especially for older people with a long-term disease or in need of regular medical care and caregiving: nursing homes for older people. The nursing homes introduced a higher standard concerning hygiene considerations, often including one- or two-person rooms with individual bathrooms [59]. The reform stipulated that the municipalities should construct new nursing homes and old people's homes according to local needs. The research sample demonstrated that these buildings adhered to the prototype for old people's homes presented in the JADA for the period of 1899–1919. In 1937, the municipality of Gentofte organized an invited architectural competition for a new old people's home (see Illustration S4) [60]. The competition brief included similar requirements as per the earlier competitions. See illustration S4 (in supplementary materials), new scan of article in 300 dpi [60].

The reform resulted in an increased number of old people's homes [59]. However, the municipalities avoided constructing nursing homes and instead sought alternative solutions [59]. Rural municipalities found that accommodating older persons in great need of assistance and caregiving in private homes was financially more advantageous than building nursing homes [59]. In 1947, an architectural competition about a new nursing home was organized in the municipality of Aarhus (see Illustration S5). Due to the Nazi occupation of Denmark in 1940, municipal investments in any types of housing for old people were postponed. See Illustration S5 (in supplementary materials), new scan of article in 300 dpi [61].

The Cooperative Prototype Promotes Homelikeness

In 1930, the cooperative organization for office employees organized an invited architectural competition for architects; however, the number of participating architects cannot be established. The assessments of first, second, and third prizes were published in the JADA. The winning proposal presented a rectangular building with one- or two-room flats with a kitchen and individual bathrooms. In 1935, the cooperative organization for journalists organized an invited competition about residential housing for their retired members. The winning proposal contained full-sized one- or two-room flats with individual kitchens and bathrooms also in rectangular-shaped buildings (see Illustration S6). Both architectural designs reflected a strong influence from functionalistic architecture inspired by modernist architecture mainly in Germany. In addition to these early competitions, two invited competitions with first, second, and third prizes were organized by housing cooperatives during the 1930s. The first one that opened in 1936 was organized by a professional organization for craftsmen. The second one opened in 1938 and was organized by a professional organization for insurance employees. See Illustration S6 (in supplementary materials), new scan of article in 300 dpi [60].

During the 1930s, a large share of people aged 65 years and older needed public support to solve their housing situation [62]. Rent took almost half of the retirement pension [62]. About 60% of the age group had their own housing, and a fifth let out a room in their private home to another old person to make ends meet [62]. The remainder of the age group leased a room in a relative's flat or house or in housing provided by charitable foundations or boarding houses [62]. A common trait for the age group was that they often ended up in buildings of the lowest housing standards. WWII prevented further architectural competitions on the cooperative form of the old people's home.

Conclusions Pertaining to Competitions 1930–1959

Contemplating old people's great need for appropriate housing, some municipalities chose to think outside the defined parameters of the social system [63]. In Copenhagen, the municipality realized that it would be far more advantageous to use part of the municipal help fund for the construction of new housing. From a legal perspective, the strategy was questionable, but it would increase the number of available dwellings for older people

cared for by municipal help funds. In 1937, the Danish Parliament legalized this use of public means [62]. A year later, this system was also introduced in Sweden [39]. So-called pensioners' homes were created, consisting of one- or two-room flats with individual kitchens and toilets and shared bathrooms and laundry rooms in the basement, identical to housing promoted by housing cooperatives. Although no architectural competition during the studied period focused on pensioners' homes, the development of this prototype for aging resembled the outcome of an architectural competition. The programming was regulated to minimize building costs and promote low rents [62].

The first pensioners' home was built in 1936: the renowned Guldberg Have in Copenhagen [63]. All the flats had central heating with individual balconies and toilets. Spaces for laundry and hygiene, bathtubs, or showers were shared by the residents in the basement. One of the five slim rectangular buildings with views in two directions had an elevator. The housing is still in use, although it was refurbished during the 1980s. Despite WWII, the expansion of pensioners' homes continued and would do so until 1956 [64]. In the municipality of Copenhagen, the total number of pensioners' homes reached 5700 during the 1950s [64]. During 1939–1945, some 63 architectural competitions were organized, but as during WWI, they mainly focused on public buildings and new building materials or building types.

In 1947, two committees were formed to evaluate the social reform of 1933 and the municipal expansion of old people's homes and new nursing homes [59]. In 1940, the ratio of older people aged 65 years and older was 7.8% of a population of 3.85 million [Statistics Denmark]. In 1949, the committees concluded that urban municipalities reasonably fulfilled the legal demand for the two types of housing for older people [59]. However, rural municipalities experienced considerable problems despite newly constructed old people's homes due to a negligent number of new nursing homes [59]. The committees revealed the municipal reluctance to expand the number of nursing homes and preference for accommodating this group of persons in need of complex caregiving in private homes [59]. To overcome the detected problems, the committees suggested state subsidies for promoting the expansion of old people's homes and nursing homes [59].

This prompted municipalities to invest in new architectural competitions on municipal prototypes for old people's homes. During the 1950s, the ratio of older people was about 9.1% of a population of 4.2 million (Statistics Denmark). Over the period of 1953–1957, three municipalities organized invited competitions concerning new old people's homes: Hoejer in 1953, Dronningborg in 1954, and Kalundborg in 1957 [35]. The competition briefs were based on the prototype for municipal old people's homes. The number of individual rooms increased while the number of shared rooms decreased. In 1956, the so-called *Small Social Reform* was passed by parliament, introducing a general improvement of public assistance and support as well as a national retirement pension for persons aged 65 years and older [46]. The JADA also revealed the outcome of the open competition about a new old people's home in Abenraa in 1957. In 1958, a new law regarding home-based elder care services was passed. It is noticeable that during the 1950s architecture for aging became a less interesting topic for architects, with few articles being published in the JADA.

### 3.2.3. Competitions 1960–1979: Architecture for Aging at a Crossroads

Over this period, the ratio of older people aged 65 years and older increased from about 10.6% of a 4.6 million population in 1960 to about 14.2% of a 5.1 million population (Statistics Denmark). In 1960, the MWSM formed a third committee to evaluate the effectiveness of care and services for the senior portion of the population. The committee was instructed to apply a future-orientated approach in its assessment of the municipal old people's homes and nursing homes [59]. National demographics anticipated an increase in older people. Despite state subsidies, the municipalities were reluctant to construct nursing homes and the number of homes increased slowly. In 1960, some 355 nursing homes with 7819 bed places existed [59]. By contrast, by the same time the number of old people's homes was around 800 with 22,000 bed places, consisting mainly of one-person rooms but also some

shared two-person rooms [59]. At some old people's homes, special hospital wards still existed with around 3400 bed places [59].

In 1962, the committee proposed the continued use of old people's homes and nursing homes, but both prototypes needed to be updated [59]. Most of the old people's homes were based on the municipal prototype of the early 20th century with uneducated staff and a high dependency on resident-shared spaces for hygiene and toilets [65]. The nursing homes, with their medically trained staff and individual bathrooms for the residents, were in better shape [65]. The committee deemed other types of special architecture for aging to be unnecessary; rather, a new model for eldercare was developed [59]. The committee imagined that the increasingly large group of older people would reside in homes within the ordinary stock of housing. The new model would focus on older persons' needs so that both caregiving and medical services could be supplied either in the ordinary home environment or in special residential care homes.

Stalemate in Cooperative and Municipal Competitions

The committee decided to expand the number of nursing homes in combination with older people continuing to live independently at their homes: 'the architectural design should to the largest extent possible be a framework for normal everyday living. ( . . . ) and that the residents should be able to lead an individual life within the flat, even though he or she participate in communal activities like any member of a community' ([66], p.11). During the 1960s, architects, the building industry, housing cooperatives, and representatives of civil administration pondered the positive and negative aspects of the two spatial prototypes for old people's homes [66], i.e., the one originating from the poor relief aid and the one coming from housing cooperatives, along with a third prototype for nursing homes closely resembling hospital wards or the municipal old people's homes.

In 1965, the ratio of older people aged 65 years and older was about 11.7% of approximately 4.8 million people, with the age group 90 years and older being especially significant (Statistics Denmark). In 1967, new guidelines for nursing homes were issued and state subsidies for refurbishing existing old people's homes were introduced [67]. The guidelines stipulated that individual flats should be at least fifteen square meters with a small hallway with closets and space for a refrigerator or a kitchenette. The flats were to have an individual bathroom of at least four square meters and were to be organized so that a person using a wheelchair or being assisted by a helper could make use of the space [67]. The subsidies resulted in several refurbishments of existing municipal old people's homes, which involved adding new wings to the existing old people's homes configured as nursing homes. Alternatively, existing old people's homes were completely rebuilt so that the new architectural configurations complied with the requirements for nursing homes.

Information about architectural competitions during the period of 1960–1979 is scarce. During the 1960s, the JADA and other documentation from the ADA reported only one architectural competition in Toender in 1964 which concerned a nursing home [35]. During the 1970s, the JADA published two articles about housing for older people; in 1979, there was a brief report about invited competitions in Heddinge, while an open architectural competition in Skagen was granted analysis in writing and drawings (see Illustration S7) [68]. See illustration S7 (in supplementary materials), new scan of article in 300 dpi [68].

Conclusions Pertaining to Competitions 1960–1979

During the period of 1960–1979, a series of changes were made to social legislation that in research regarding social welfare fall under the concept of the Third Social Reform. Changes were made to laws on public assistance and public health insurance as well as on daily allowances in the case of sickness or childbirth. The reforms meant that the municipalities assumed the key roles in the realization of social care and support for their citizens [46]. In 1970, the ratio of people aged 65 years and older was 11.9%, while the population was around 4.9 million people (Statistics Denmark). The decade was characterized by a thorough questioning of existing beliefs and ideals in architecture for the

dependent and frail aging process. The social act was replaced by a new law regarding social services in 1976. Other changes were made to public health insurance [69,70]. The reform led to a redefinition of the nomenclature for existing architecture [71]. Given the various building types for housing older people, the reform made the following clarifications toward understanding the different forms of housing for senior Danes:

- Municipal homes, which were originally named old people's homes or older denominations, were renamed residential care homes.
- Municipal pensioners' homes and the old people's homes of housing cooperatives were renamed 'housing for elderly people'.
- In addition to these updated existing concepts, a new prototype for aging was created, i.e., sheltered housing, normally comprising 1–2 room flats often integrated into refurbished older buildings previously used as municipal nursing homes or old people's homes.

The majority of these types of homes were under municipal control, but private initiatives for alternative forms of housing also started to appear. These senior co-housing residences normally comprised a block of ordinary flats in which residents shared a feeling of belonging together based on mutual interests [69,70]. In 1970, the government appointed a special committee, the Old Age Commission (hereafter referred to as OAC) to investigate senior Danes' living conditions [70,72].

### 3.2.4. Competitions 1980–2012: Appropriate Space for Aging Reinvented

During the 1980s, it was clear that the condensed space and large number of shared spaces for hygiene, i.e., bathrooms, laundry rooms, storage, and toilets, constituted a severe problem in existing architecture for older people of the early 20th century [62,73]. The strict focus on building costs and low rents had limited the level of flexibility in these forms of housing to meet future user needs, e.g., as pensioner's homes. This problem also concerned vertical communication due to narrow stairways and few elevators or elevators that could not be used by people who used mobility devices (walkers, wheelchairs). Due to the low housing standards in architecture for older people, younger users in individual need of social support started to obtain flats in these types of housing. In 1989, one quarter of the residents in pensioners' homes were younger than 65 years of age [62].

In 1982, the OAC concluded that housing within the ordinary stock of housing should be the ideal architecture for aging considering any aging process and regardless of individual needs for care and caregiving [74,75]. The OAC deemed the institution-like environment to be unsuitable for aging and promoted ordinary residential architecture—the homelike setting—as the envisioned space in which to grow old. The home environment could be adjusted to personal needs so that the individual aging process was always in harmony with self-image. The OAC formulated three principles for appropriate eldercare for frail older persons: continuity, self-determination, and individualized caregiving [74]. Consequently, deinstitutionalization and expanded access to home-care services were to characterize modern eldercare.

In 1987, the OAC initiated a longitudinal study on Danes' views on appropriate housing for aging: the Danish Longitudinal Future Study [72,76]. Since then, 460 people in three age groups (40–44, 50–54, and 60–64 years) have been interviewed on two more occasions, in 1997 and 2002 [72]. The interviews confirmed that the home environment, appropriated during a lifetime, was the ideal habitat for aging. In 1988, the social act was reformed again to include subsidies for refurbishing existing old people's homes and pensioners' homes and to block the further expansion of nursing homes until 1993 [62]. The reform put a permanent end to the further reuse of the discernable four prototypes for aging from the first half of the 20th century, i.e., (1) the municipal old people's home, (2) the cooperative old people's home, (3) the nursing home, and (4) the pensioners' home. Instead, these prototypes merged to form two new prototypes for appropriate aging:

- Residential care homes in which the architectural design took into consideration high demands on physical accessibility, flexibility, and usability for a wide range of users,

including people with disabilities. Thus, the physical environment allowed residents to continue to lead an independent life in small flats with access to communal space for activities, meals, and social contact. In addition, the homes were integrated into a safe and secure care environment with 24-h caregiving.

- Residential housing for senior Danes in which the architectural design incorporated a high level of physical accessibility and flexibility so that a wide range of users, including people with disabilities, could lead a comfortable life, but also allowing for adequate work–environmental requirements for home-based eldercare services. This housing was to be situated near buildings with 24-h caregiving.

In 1995, the Danish Parliament passed another reform that concerned the existing nursing homes. New spatial requirements were introduced with a higher level of physical accessibility required than for other types of buildings, thereby focusing on the potential needs of older persons or people with disabilities, e.g., improved bathroom designs of about seven square meters [77]. The restructuring of nursing homes resulted in a new type of architectural space that respected the early OAC principles. This new type of nursing home was often labeled 'live and reside milieu' [65]. The main characteristics of this architectural design were two-room flats in an open configuration with access to communal space for restaurants and various activities. The new nursing homes continued to be extensions and refurbishments of existing housing for older people, but they could also constitute the first step in the development of a new area for a housing settlement. The latter use often resulted in an off-side location compared with other residential architecture.

In 1997, about 58% of people aged between 70 and 74 years were interested in moving to new LRM facilities [78]. However, once installed, many older persons felt old since aging became much more apparent in areas with only one age group in comparison with a flat in the ordinary stock of residential space [78]. Consequently, in 2004, interest in moving to such milieus had dropped to 38% for this age group [78]. This drop correlated with the segregating effect of the off-side location of these facilities. In 1998, the social act was once again reformed by the Service Act. A social–liberal principle was reintroduced that emphasized the individual responsibility of the receiver of a social service to regain independence from public support. The new law divided public assistance into three levels depending on the receiver's needs, ranging from advisory help for various matters, access to the municipal social framework, to financial support in case of special health or social problems [79]. In 2005, the ratio of older people aged 65 and older was 15.0% of a 5.4 million population (Statistics Denmark). The age group 85 years and older in particular increased considerably during the 1990s and the new millennium.

Abandoning Old Prototypes—Renewal and Innovation

During the 1980s, the number of architectural competitions that focused on architecture for aging started to increase with a total of 14 competitions, about 90% having a municipal organizer. An analysis of the competitions suggests that the 1980s was a turning point in the development of Danish architecture for aging. The early competitions during the period of 1980–1984, about eight competitions, were mainly orientated towards renewing existing nursing homes, i.e., older prototypes for nursing homes in need of refurbishment or other prototypes undergoing refurbishment to become nursing homes according to the physical requirements for nursing homes that had been introduced in 1967. In addition, the refurbishments included updates according to the OAC principles for continued independent living within a homelike setting with access to individual 24-h caregiving (see Illustration S7).

The focus of the competitions started to shift for those organized from 1985 to 1989, comprising nine competitions. They focused on ordinary housing for elderly people in general. The updated understanding of residential housing for senior Danes promoted large areas with various buildings for older people, often called elderly centers, which merged home-like and small-scale residential architecture with the structural advantages of large-scale institutions (see Illustration S8). Architectural competitions for the period of

1985–1989 promoted two-room flats with a separate space for sleeping that were intended for a single person or couple. The communal space included space for restaurants, social and sport activities, and space for physiotherapy. See illustration S8 (in supplementary materials), Old snapshots attached [80].

The following decade of 1990–1999 registered a record number of 25 architectural competitions. The municipalities organized 15 competitions, while 6 competitions were co-organized between a municipality, a housing cooperative, and a player from the civil administration (a county or a ministry). Four competitions were organized by a housing cooperative. At the beginning of the decade, the competitions concerned the refurbishment of existing old people's homes, nursing homes, or pensioners' homes, often in combination with new buildings. The design tasks concentrated on different types of housing for older people on a single plot which were often hinted at in the name of the competition, e.g., centers for older people, eldercare centers, and housing for older people with residential care homes. The national debate on appropriate architecture for aging continued the spatial exploration of appropriate architecture for aging. For the period 2000–2012, 22 architectural competitions were organized, with approximately two per year. Housing cooperatives organized two competitions, while the majority had a municipal organizer. These competitions focused on the new type of nursing home called LRM housing (see illustration S9). Illustration S9 (in supplementary materials), floorplan courtesy Cubo Architects to use the plan and cited source page 12 [81].

Conclusions Pertaining to Competitions 1980–2012

In research on social welfare, the reforms of 1988–1998 are seen as part of a *Fourth Social Reform*. Subsidies for cooperative, municipal, or private senior co-housing projects for older people were also introduced [73]. Given the strong common interest in the conditions required for aging well and influential organizations in defense of older people's rights, the matter of appropriate housing for aging entered the national political agenda. In 2004, the government allocated state funding to a four-year research project about appropriate housing for aging called 'Well-being and housing'. The research was realized over the period of 2004–2008 and was led by the Center for Applied Health Services Research (CAST) at the University of Southern Denmark in Odense and the Department for architecture and design at the Aalborg University in Aalborg [28].

The research task was to shed light on the relationship between architectural design and old persons' needs [30]. Thus, the research focused on older persons' sensations of connecting to a place and experiencing safety in a home and in a familiar environment [30]. In 2012, a set of preliminary guidelines for the homelike design of new residential care homes was presented. From 2013, they have annexed or even replaced competition briefs in later architectural competitions. Architecture for aging in the 21st century is focused on the capacity of the architectural design to instill the solitude of a safe and secure home environment—homelikeness—which allows for aging comfortably despite a long-term health condition.

## 4. Discussion

This study had a twofold aim, with a transdisciplinary approach to studying architectural competitions on architecture for aging during the period of 1899–2012 while also exploring potential links between the competitions and changes in social legislation. With this double objective, the intention was to retrace the generating images that were activated during the evolution of housing for older people in the modern Danish welfare state. Akin to research on architectural competitions in neighboring countries such as Norway and Sweden [26,27], this study demonstrates that investment in competitions originated from the introduction of a small age pension in the early 1890s. Other reforms followed that called for new architectural competitions. The full research sample suggests that competitions in architecture became a socio-political instrument to define architectural perimeters that were in line with social ambitions.

The reform of 1890–1891 was based on a cultural belief that aging could be viewed in two ways: as an undignified aging process with extensive dependency on society, or as a dignified aging process with maintained independency but in a vulnerable position. With this distinction in mind, the research shows that architectural competitions contained a spatial discussion about which type of architecture matched the two aging processes. The municipalities, which already administered poor relief aid to older persons, relied on an institution-like architecture that made use of the rationalizing aspect of architecture to regulate a problem. Some ten years later, housing cooperatives affiliated with organizations for professional groups initiated competitions that promoted a homelike architecture that explored architecture as a supportive environment for existence.

During the century that followed, the spatial discussion following social reforms made use of these two main opposing prototypes to define the parameters for today's recommended architectural designs for architecture for aging, especially for frail aging. As shown in Figure 3, the competitions demonstrate a tension between architecture as structures for solving a problem with society as the main agent and architecture as a design practice to meet the personal needs of a smaller group of users. Seen as an entity, the architectural competitions of 1899–2012 promoted a new understanding of architecture as a locus of spatial experiences that enrich living in the later stages of life. As time went on, the original spatial prototypes for old people's homes and nursing homes were no longer viable pieces of architecture to address aging in the modern welfare society.

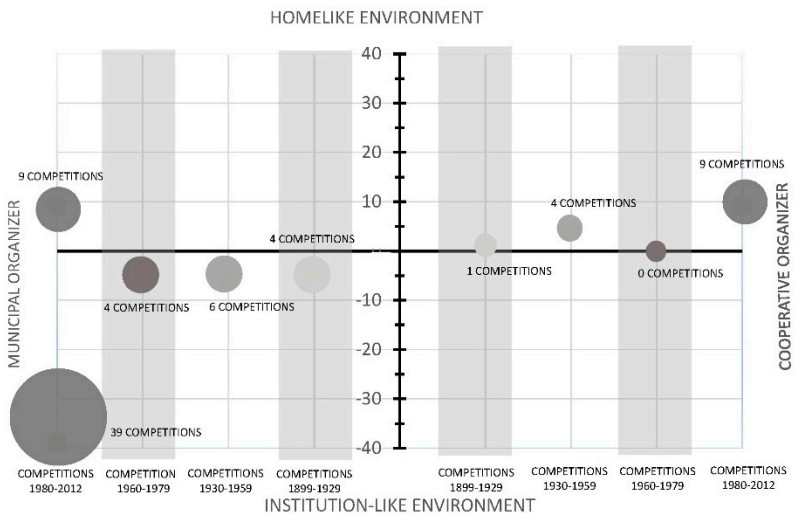

**Figure 3.** Distribution of the research sample in a matrix that relates type of architecture to type of organizer. Increased focus on the individual needs of the aging person has come to promote a homelike architecture over an institution-like architecture (figure by author).

Using architectural competitions as a corpus for understanding a centennial evolution of both architecture and social welfare for dependent and frail older people implies limitations, and conclusions are subject to interpretations and inference. However, this study demonstrates that architecture for aging was an important social matter that involved discussions between decision-makers, i.e., politicians, and advisors, i.e., architects, care planners, and economists. These discussions were particularly intensive during the period of 1960–1980 since conflicting requirements were approximated into solutions that were deemed to be feasible types of architecture for aging [26,27]. The rich production of white papers suggests that architectural competitions on social matters such as architecture for aging must be considered from an architectural point of view that is anchored in societal context.

The sample of architectural competitions from the period of 1899–2012 reveals a professional condensation of views on the aging process and corresponding ideal environments that resulted in an architectural statement that compressed aesthetics, ethics, ideology, and

rational reasoning into spatial prototypes open to further discussions and refinement. This study suggests that architectural competitions on architecture for aging were part of an evolution towards a welfare regime with a high level of immunization for citizens against the effects of societal changes and towards a system of social entitlements. In modern Denmark, architecture for aging is dependent on the socio-political decisions that define its perimeters. Currently, architecture for aging in the 21st century is focused on the potential of architecture to convey homelike features—homelikeness has become a design criterion.

## 5. Conclusions

With a twofold aim and a transdisciplinary approach, this study explored architectural competitions on architecture for aging in Denmark during the period of 1899–2012 and their potential linkage to changes in national social legislation. The sources for this research had a transdisciplinary nature, consisting of publications for the architecture profession and architectural research; papers and publications from research on social welfare; and white papers from civil administration on a national level.

This study demonstrated that architectural competitions were used as a socio-political instrument for defining the perimeters of social reforms. The competitions produced spatial prototypes that described either an institution-like environment or a homelike setting. During a centennial process, these prototypes generated development towards contemporaneous guidelines that promoted homelike aspects in architecture for aging.

**Supplementary Materials:** The following supporting information can be downloaded at: https://www.mdpi.com/article/10.3390/architecture3010005/s1. Illustration S1. Floorplan of the old people's home in Aarhus and exterior views, architect V. Schmidt [51], © Danish Association of Architects [Akademisk Arkitektforening]. Illustration S2. Old people's home in Frederiksberg, competition proposal 1919, architect S.C. Larsen [54], © Danish Association of Architects [Akademisk Arkitektforening]. Illustration S3. One of four winning competition proposals 'Concentric circles' by architect S. Lemche [57], © Danish Association of Architects [Akademisk Arkitektforening]. Illustration S4. Winning proposal by architects Langkilde and Jensen for the architectural competition about an old people's home in the municipality of Gentofte in 1938 [60], © Danish Association of Architects [Akademisk Arkitektforening]. Illustration S5. Winning proposal for a nursing home in the municipality of Aarhus in 1947, architect Salling Mortensen [61], © Danish Association of Architects [Akademisk Arkitektforening]. Illustration S6. Drawings and exterior view of the realized building in 1938 for the architectural competition about a cooperative old people's home organized by the Association for Journalists in 1935, architect W. Lauritzen, [60], © Danish Association of Architects [Akademisk Arkitektforening]. Illustration S7. Winning proposal in the architectural competition about a nursing home in the municipality of Skagen, architects Justensen and Kastow [68], © Danish Association of Architects [Akademisk Arkitektforening]. Illustration S8. Winning proposal for elder center with housing and services in the municipality of Rindkoeping, architectural competition in 1996, architects Andersen and Hvid [80], © Danish Association of Architects [Akademisk Arkitektforening]. Illustration S9. Winning proposal in architectural competition organized by the municipality of Frederiksberg in 2011 and focusing on future-oriented housing for older people, architect Cubo Arkitekter A/S, Copenhagen. Upper floorplan © Cubo Arkitekter A/S, Copenhagen, axonometric view © Frederiksbergs Kommune [81].

**Funding:** This research received no external funding.

**Data Availability Statement:** Not applicable.

**Acknowledgments:** The first part of this study was carried out in 2012–2013 at the Danish Building Research Institute, SBi, at Aalborg University, AAU, in Copenhagen, Denmark. My gratitude goes to Hedvig Vestergaard at the Danish Building Institute, Copenhagen for her continuous inspirational input to my work. The initial study was further elaborated during 2013–2015 to prepare for a first publication of the results in Danish in an anthology on housing for senior Danes. I would also like to thank Per H. Jensen at the Aalborg University in Aalborg and Tine Rostgaard at VIVE, The Danish Center for Social Science Research, Copenhagen for their support when finalizing a first draft of the study in Danish. The conversion of the Danish study into a publication necessitated further research over the period of 2015–2019.

**Conflicts of Interest:** The authors declare no conflict of interest.

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
