# Peer review of "Architectural Competitions on Aging in Denmark Spatial Prototypes to Achieve Homelikeness 1899–2012"

_2673-8945, doi:10.3390/architecture3010005_

Round 1

Reviewer 1 Report

This study covers a very broad historical period, more than a century in fact. It seeks to establish a taxonomy of architectural typologies and to relate them to phases of a social policy of housing the elderly in Sweden. The study is very interesting. It draws on archival analyses that the author(s) conducted in 2012 and 2013.

It is imperative to specify the origin of the documentary sources as well as the holders of the publication rights. Figure 1 lists 4 bibliographic references that suggest that the diagram was produced by other authors. It is necessary to specify if the figure 1 is free of rights. The same goes for the illustrations that refer to bibliographic references without mentioning the pagination. The general quality of the illustrations is rather low, with illustrations 4 and 8 being blurred and needing to be improved for better readability. Figure 2 is the most important and probably the main scientific contribution of the article. Its legend should be amplified and more detailed in order to highlight the contribution to knowledge and better explain why some prototypes are found together in the 4th social reform.

In the conclusion the authors consider that the competitions served as a filter to understand the evolution of architecture. It seems to me that the notion of filter is ill-fitted to the methodological approach since it is clear that the competitions were used as corpora rather than as filters. The almost perfect division between typologies and periods of social reform should be more carefully considered in the introduction and at least in the conclusion, as one would expect greater tensions and variations within each period. How did the authors ensure that they did not project their own preconceptions in their methodological approach?

Since the authors acknowledge that the complexity of the competition documents consists of "condensation", how did they ensure the accuracy of the analyses aimed at untangling the threads of these complexities?

The paper still lacks certain methodological elements to appreciate the accuracy of the analyses.

One might even wish that the study covered only the last two phases of social change in order to restrict the scope of the corpus by allowing a more detailed exposition of typological changes. One century is certainly too broad and the paper is not synthetic enough in its present form. Shouldn't the tension between the two categories (home like, institution like) appear in the introduction as a kind of methodological duality that the research tends to nuance as it progresses in the examination of empirical data? 

Author Response

This study covers a very broad historical period, more than a century in fact. It seeks to establish a taxonomy of architectural typologies and to relate them to phases of a social policy of housing the elderly in Sweden.

The study is focused on Danish architecture, with some relevance for Sweden regarding housing frail older persons

The study is very interesting. It draws on archival analyses that the author(s) conducted in 2012 and 2013.

Details on archival research with a trans-disciplinary approach have been included in section 2.

It is imperative to specify the origin of the documentary sources as well as the holders of the publication rights.

This issue has been rekindled given that Nordic copyrights seem more generous than copyrights used by English-speaking journals. Normally, according to Nordic principles images and drawings in publications can be re-used in articles with scientific purpose. However, the matter is under investigation, resulting either in a photograph from the JADA or a new drawing/ illustration based on the JADA material.

Figure 1 lists 4 bibliographic references that suggest that the diagram was produced by other authors. It is necessary to specify if the figure 1 is free of rights.

This has been corrected – figure 1 is made by the author by use of the full research material.

The same goes for the illustrations that refer to bibliographic references without mentioning the pagination.

To the extent possible, this has been addressed, but the illustrations are also checked according to recent copyright legislation.

The general quality of the illustrations is rather low, with illustrations 4 and 8 being blurred and needing to be improved for better readability.

This will be addressed, either with new illustrations based on the original publication or as redrawn material using CAD technique. This is related to the question of copyrights.

Figure 2 is the most important and probably the main scientific contribution of the article. Its legend should be amplified and more detailed in order to highlight the contribution to knowledge and better explain why some prototypes are found together in the 4th social reform.

This has been addressed to the extent possible.

In the conclusion the authors consider that the competitions served as a filter to understand the evolution of architecture. It seems to me that the notion of filter is ill-fitted to the methodological approach since it is clear that the competitions were used as corpora rather than as filters. The almost perfect division between typologies and periods of social reform should be more carefully considered in the introduction and at least in the conclusion, as one would expect greater tensions and variations within each period.

Noted, perhaps corpus is a better word than filter, so this word is used in the conclusion.

How did the authors ensure that they did not project their own preconceptions in their methodological approach?

The codex of ADA was used as a referential framework.

Since the authors acknowledge that the complexity of the competition documents consists of "condensation", how did they ensure the accuracy of the analyses aimed at untangling the threads of these complexities?

This was overcome by consulting white papers of the civil Danish administration that can be found in the registry of the Danish parliament or in the national library data-base.

The paper still lacks certain methodological elements to appreciate the accuracy of the analyses.

Some more details have been added to the second section.

One might even wish that the study covered only the last two phases of social change in order to restrict the scope of the corpus by allowing a more detailed exposition of typological changes.

This option was initially considered, but as the paper show, in the national discussion spatial prototypes were used based on their functionality rather than on their chronology.

One century is certainly too broad and the paper is not synthetic enough in its present form.

We are aware of the long period, but the problem is that the spatial prototypes were used regardless of chronology.

Shouldn't the tension between the two categories (home like, institution like) appear in the introduction as a kind of methodological duality that the research tends to nuance as it progresses in the examination of empirical data? 

This has been addressed as suggested in the introduction.

Reviewer 2 Report

The study intends to retrace how architectonic visions of housing with caregiving for older people have evolved in the Danish welfare society. It provides an extensive overview of the socio-political changes that impacted the spatial quality of the building prototypes dedicated to this marginalised group. However, in this current format, the article is quite descriptive. The research methodology is primarily literature-based, which is good, but some of the identified aspects that the author mentioned needed to be more successfully managed to establish in the entire paper. Like the ideals of an aesthetic, it can hardly be judged with the current text and the type of quality of the images the author used in the article. Most plans and elevations could be of better quality to understand and interpret the arguments meaningfully. It would be better if necessary to redraw the images. Apart from that, the article is well written and will be a good historical research product for architects in the Danish context.  

Author Response

The study intends to retrace how architectonic visions of housing with caregiving for older people have evolved in the Danish welfare society. It provides an extensive overview of the socio-political changes that impacted the spatial quality of the building prototypes dedicated to this marginalised group. However, in this current format, the article is quite descriptive.

The present version has been addressed, especially by removing details that only are familiar to Danish citizens.

The research methodology is primarily literature-based, which is good, but some of the identified aspects that the author mentioned needed to be more successfully managed to establish in the entire paper.

The introduction has been rewritten and homelike versus institutionlike have been introduced.

Like the ideals of an aesthetic, it can hardly be judged with the current text and the type of quality of the images the author used in the article.

This matter had been addressed by investigating copyright issues. Danish copyrights allow for using images and drawings from other publications in scientific papers, but obviously this is not the same in English copyrights.

Most plans and elevations could be of better quality to understand and interpret the arguments meaningfully.

This matter had been addressed by investigating copyright issues. Danish copyrights allow for using images and drawings from other publications in scientific papers, but obviously this is not the same in English copyrights.

It would be better if necessary to redraw the images.

This matter had been addressed by investigating copyright issues. Danish copyrights allow for using images and drawings from other publications in scientific papers, but obviously this is not the same in English copyrights. This might be a potential outcome to redraw all the illustrative material.

Apart from that, the article is well written and will be a good historical research product for architects in the Danish context.  

Thank you very much for the encouragement!

Reviewer 3 Report

The abstract should be rewritten in order to be aligned with the journal, which We strongly encourages authors to use the following structure: Background, placing the question addressed in a broad context and highlight the purpose of the study; Methods, briefly describing the main methods or treatments applied; Results, summarizing the main findings; Conclusions, indicating the main conclusions or interpretations.
The reference style does not follow the style of the journal.
In the introduction, authors should upgrade their statistics (for instance, those coming from 2014 regarding the proportion of people aged 65 years).
When classifying type of housings, four categories (line 30) with three numbered indexes?
Regarding architectural designs and user needs, can the references be reviewed? Why is not any research in last 10 years?
Authors should stress which is the gap to be solved, which is the novelty (for highlighting their contribution).
In relation with the aim and purpose, which kind of analysis is going to be performed?
What about the (private) architectural work that is not subject to competitions and that also meets these needs?
The
critical analysis process must be better described and justified because it is only referenced.
This type of analyses should consider both the specifications of the organisation promoting the competition and the results of the contestants, as well as the motives of the promoting organisations in favouring some proposals over others.
What is the evolution of both the objectives to be scored and their weighting?
Explicitly, a typological analysis of the spaces and their relationships should be considered, based on the study of the 76 competitions collected.
Some images have lost resolution. It is recommended to take care of these figures.

Author Response

The abstract should be rewritten in order to be aligned with the journal, which We strongly encourages authors to use the following structure: Background, placing the question addressed in a broad context and highlight the purpose of the study; Methods, briefly describing the main methods or treatments applied; Results, summarizing the main findings; Conclusions, indicating the main conclusions or interpretations.

The introduction is rewritten and somewhat shortened, with a clearer focus on architectural competitions.

The reference style does not follow the style of the journal.

This has been addressed and changed to follow journal standard.

In the introduction, authors should upgrade their statistics (for instance, those coming from 2014 regarding the proportion of people aged 65 years).

This information has been removed completely to concentrate on the theme of architectural competitions.

When classifying type of housings, four categories (line 30) with three numbered indexes?

This information has been removed so that focus is solely on architectural competitions.

Regarding architectural designs and user needs, can the references be reviewed? Why is not any research in last 10 years?

New developments in architecture for the frail ageing process after 2013 is subject to another research project.

Authors should stress which is the gap to be solved, which is the novelty (for highlighting their contribution).

This is addressed somewhat more in the new version.

In relation with the aim and purpose, which kind of analysis is going to be performed

The working hypothesis has been rewritten to highlight the type of research: The working hypothesis was that socio-political ideals programmed architectural competitions. Therefore, the submitted proposals encapsulated veins of the ongoing socio-political discussion about appropriate space for ageing [31, 36]. When realized, these winning proposals constituted a bank of referential prototypes for housing for frail older persons that stayed active for a long period, creating either imaginary cul-de-sacs or openings to new ideas for the corpus of architects [12, 37].

What about the (private) architectural work that is not subject to competitions and that also meets these needs?

The research material has only concentrated on architectural competitions, not individual presentations of other projects for residential care.

The critical analysis process must be better described and justified because it is only referenced.

This type of analyses should consider both the specifications of the organisation promoting the competition and the results of the contestants, as well as the motives of the promoting organisations in favouring some proposals over others. What is the evolution of both the objectives to be scored and their weighting?

The research material mainly promotes the winning proposal not the contestants’ proposals other than second and third prize winners.

Explicitly, a typological analysis of the spaces and their relationships should be considered, based on the study of the 76 competitions collected. Some images have lost resolution. It is recommended to take care of these figures.

This has been addressed when looking at copyright issues.

Reviewer 4 Report

The article is long and tedious to read. I recommend an in-depth summary of the text in order to make its reading more enjoyable and synthetic, since its conclusions are not far-reaching enough to extend the text so much.

Author Response

The article is long and tedious to read. I recommend an in-depth summary of the text in order to make its reading more enjoyable and synthetic, since its conclusions are not far-reaching enough to extend the text so much.

We note the comment and have made language revisions with the help of a Native-English speaking person, but keeping the overall format.

Round 2

Reviewer 3 Report

The paper has been improved. However, there are some questions that should be taken into account.

The case study as a methodology must be stressed explaining how your research is enriched if other methods are discarded.

The title includes 2 considerations: First, a specific period of time has been selected: 1899-2013. The origin and the end of it must be justified (to include last ten years in another project is not enough). Second, the review should lead to a series of spatial prototypes but the abstract does not reflect this.

In the abstract, sentences such as "the true essence of home in architecture for ageing" should be avoided because of lack of explanation. The abstract should be aligned with the title.

In the introduction, the discarding of the last 10 years should be justified.

The research focuses on architectural competitions but it does not include the private enterprise. The percentage of number of habitational solutions for older people that come from public tenders or private direct commissions. 

The title does not consider the competition as a tool for (because does not try to prove it).

The introduction should include the objetives of the research. In addition, the introduction could include in its last paragraph the description of following sections.

Representativeness of JADA must be provided.

Which secondary literature, study visits and white papers were consulted? This is not enough.

The evolution of the ratio older people/people and its translation to the number of competitions is not considered.

The review of the Illustrations 1, 2, 3, 4, 5, 6, 7, 8, 9 cannot be deployed.

They are relevant for the research, so this is an inconvenience.

The conclusions are too superficial, without entering into a topological, spatial, material, constructive classification or evolution. No limitations are pointed out nor future research suggested.

Author Response

Dear reviewer, 

many thanks for constructive comments that have been helpful in improving the paper. The following revisions of the text have been undertaken:

  1. methodology - this part of the text has been completely rewritten so that the full pursuit of research can be retraced, case study methodology with a mixed method approach, primarily archival research methods.
  2. the time period 1899-2013 has been clarified: competitions before 2013 follow the normal procedure for architecture competitions but in Denmark since 2013, the procedure for competitions about architecture for ageing have changed since national requirements for social architecture have been defined. This is new both for architectural competitions per se and competitions on architecture for ageing. 
  3. Abstract with spatial prototypes - the abstract has been completely reworked and rewritten. The new abstract now gives information about research objectives, methods and conclusions. 
  4. title and abstract: the title has been changed to reflect the new angle for the study, consequently, the abstract follows this approach. The use of 'true essence' has been modified in the sense that true has been removed. potentially, there is a language issue at the bottom of this use, but we agree the reviewer that true essence is unclear.
  5. The introduction has been rewritten like the full paper, however, the track-changes-mode has not worked throughout the text, only in the opening parts of the text. The discarding of ten years from 2013 until today is explained in the beginning of the result section - competitions prior to this date are different than the ones after this date due to the use of a national list of requirements. 
  6. The focus for architectural competitions - apparently, a language issue has created a problem here. The players involved in the competitions that were found in the professional journal for architects were representatives of the Danish civil administration, i.e., ministries, counties and largely municipalities. However, other players were also involved representing real estate owners but not building companies. Typical for Denmark, housing is owned by housing cooperatives that represent different interests that can be associated with professional affiliation, saving, or funds. To simplify, housing companies have been changed to housing cooperatives. The number of competitions per player has been included
  7. the title has been changed once again, but highlighting spatial prototypes. That the competitions have been an instrument in the development of the Danish social legislation has been included as a conclusion in part 5.
  8. The introduction has been rewritten and now states objectives: The aim of the present study was to pursue similar systematic research of Danish architectural competitions to retrace architectonic visions for housing for frail older people during the evolution of the modern welfare state. The working hypothesis was two-folded: firstly, that socio-political ideals from different stages in the formation of the Danish social act would be reflected in architectural competitions as in the neighboring countries. Secondly, that spatial prototypes for ageing would emerge. 
  9. The representiveness of the JADA has been explained by use of new references and explanation about competitions in the European context: As in many European countries, ever since the mid-19th century, the Nordic professional organizations that are open for trained architects are in control of the architecture competition as an architectural quality instrument. Following the European beaux-arts tradition, competing in architecture is more than a competition, it is a quest for genius. Hence, winning a competition is not only landing a new commission, within the architecture profession, it is also a recognition of possessing outstanding skills in architecture: to quote the French architectural theorist Quatremère de Quincy competing in architecture is about ‘suppress the ignoramuses’ and ‘find the ultimate spatial solution’ [87, 88, 89]. 
  10. secondary literature has been explained further in the completely rewritten section on material and methods: 

    Maps, mainly supplied by Google Earth, and addresses were studied to establish whether the competition had resulted in a new building or settlement. Public records and websites, open searches with the Google Search motor and supplied by the website, were consulted to deduce current activities, and prepare for study visits. The matter of ageing could also be explored by consulting the sources that the JADA indicated in the national library database (www.kb.dk). Most of this material was accessible and possible to scrutinize at the Royal Library in Copenhagen.

  11. the ratio of older people aged 65 years and older has been included in the text based on statistics from Statistics Denmark, traceable back to mid 19th century. Where suitable, an estimate has been included to give an indication about demographics. 
  12. Still, working on the illustrations - according to Danish copyright law it is possible to include copies of the original image from the publication. The JADA has been asked about supplying a written statement that allow us to use these copies, however, they have not yet responded. Practical reasons has made it impossible to get access to the material shortly before Christmas and during Christmas. But we concur, the illustrations and the drawings would give additional information to our arguments about the spatial typology. To redraw the material with modern CAD technique would be a clear problem for the validity of the study. 
  13. the section 5. conclusion has been completely rewritten and includes the conclusions that can be extracted from the study on a comprehensive level - competitions used as a tool for defining space for ageing, producing prototypes to reason in a rational tone about architecture for ageing. 

Finally, we are very grateful for the constructive comments and it is our hope that we have managed to solve most of them. 

Round 3

Reviewer 3 Report

The paper has been improved again.

However, some minor concerns remain at present.

For instance, failure to provide the illustrations should delay the publication.

On the other hand, I disagree on the presentation of theorems. I think that a series of hypotheses should have been presented, which would then be proven by the method.

Author Response

Dear reviewer 3,

thank you very much for constructive comments on how to edit the manuscript. Revisions have been made to the introduction in order to make way for a new opening of the result section. The introductory part of this section ends with the theorem - two types of ageing, the undignified versus the dignified - that the competition sample to our minds seem to describe - municipal space versus cooperative space. Hopefully, this is an improvement in the direction that you suggest. 

For this revision, it has been possible to update the illustrations, which have been flat-scanned by the Royal Library for Architecture and Arts in 300 dpi. We include full-sized scans that in Danish context could be cropped to fit the paper. Just illustration 8 has presented some problems since the issue was missing at the library. We will consult another library to get hold of a new flat-scanned material. The included material is snapshots made with a cell during research. 

The presentation of the foundation for the theorem has been supplied by rewriting the introductory part of the result section as previously described. 

Many thanks for having the patience to read our manuscript. Looking forward to your assessment.